# The inclusion of Amazon mangroves in Brazil's REDD+ program

Angelo F. Bernardino [1] ✉, Ana Carolina A. Mazzuco [1,9], Rodolfo F. Costa[2], Fernanda Souza[3], Margaret A. Owuor[4,5], Gabriel N. Nobrega[6], Christian J. Sanders [7], Tiago O. Ferreira [2] & J. Boone Kauffman[8]

The Legal Amazon of Brazil holds vast mangrove forests, but a lack of awareness of their value has prevented their inclusion into results-based payments established by the United Nations Framework Convention on Climate Change. Based on an inventory from over 190 forest plots in Amazon mangroves, we estimate total ecosystem carbon stocks of $468 \pm 67$ Megagrams (Mg) ha$^{-1}$; which are significantly higher than Brazilian upland biomes currently included into national carbon offset financing. Conversion of mangroves results in potential emissions of 1228 Mg $CO_2$e ha$^{-1}$, which are 3-fold higher than land use emissions from conversion of the Amazon rainforest. Our work provides the foundation for the inclusion of mangroves in Brazil's intended Nationally Determined Contribution, and here we show that halting mangrove deforestation in the Legal Amazon would generate avoided emissions of $0.9 \pm 0.3$ Teragrams (Tg) $CO_2$e yr$^{-1}$; which is equivalent to the annual carbon accumulation in 82,400 ha of secondary forests.

Brazil has one of the largest global greenhouse gases (GHG) emissions largely due to deforestation and degradation of forests, and conversion to croplands and cattle pastures[1]. Decreasing deforestation is a government priority to achieve Brazil's Intended Nationally Determined Contributions (INDC), which set a 100% emissions reduction target from the current deforestation under the Land Use, Land Use Change, and Forestry (LULCCF) emissions of -947 Tg$CO_2$e by the year of 2030[2]. The potential avoided deforestation could be incorporated into voluntary carbon credits to finance forest conservation through the REDD+ (Reducing Emissions from Deforestation and forest Degradation) initiative, although double-counting of carbon credits among REDD+ projects and those incorporated in INDC's are being discussed by the United Nations Framework Convention on Climate Change (UNFCCC)[3,4]. The ambitious Brazilian REDD+ forest reference levels (FREL) for the forestry sector raised major global interest as it

may support the protection of pristine upland forests with large international funding[4]. Given the raising global interest in the voluntary carbon markets, funding through REDD+ programs could raise billions of dollars and potentially help cutting the annual deforestation rates of -1 Mha in the Amazon biome[5].

The inclusion of mangrove forests and other coastal wetlands in Brazil's REDD+ projects would bring ecological and socioeconomic benefits. These wetland forests are among the largest carbon stocks of any forest ecosystem and can be significant sources of GHG once disturbed[6–8]. The loss of mangrove forests have been shown to release massive amounts of carbon that have been sequestered for centuries in deep soils and even in biomass[9]. Therefore, the carbon stocks in mangrove forests are considered to be of great value for global climate change mitigation strategies[9–12], and Brazil is likely missing a large natural coastal carbon sink on the Amazon coast. The absence of

[1]Departamento de Oceanografia, Universidade Federal do Espirito Santo, Av. Fernando Ferrari, 514, Goiabeiras, Vitória-ES 29075-910, Brazil. [2]Department of Soil Science, Luiz de Queiroz College of Agriculture, University of Sao Paulo, Piracicaba, SP, Brazil. [3]Instituto Chico Mendes de Conservação da Biodiversidade, Ministério do Meio Ambiente, Macapá, AP, Brazil. [4]Wyss Academy for Nature at the University of Bern, Bern, Switzerland. [5]Institute of Ecology and Evolution, University of Bern, Bern, Switzerland. [6]Departamento de Ciências do Solo, Universidade Federal do Ceará, Fortaleza, CE, Brazil. [7]National Marine Science Centre, Southern Cross University, Coffs Harbour, NSW 2540, Australia. [8]Department of Fisheires, Wildlife, and Conservation Sciences, Oregon State University, Corvallis, OR, USA. [9]Present address: UNESCO/IOC Project Office for IODE, Flanders Marine Institute, InnovOcean Campus, Oostende, Belgium. ✉ e-mail: angelo.bernardino@ufes.br

mangroves in Brazil's climate mitigation framework could be reasoned by their limited extent of approx. 1 Mha[13]; when compared to the large upland biomes (>850 Mha), and a lack of recognition of their high carbon stocks when compared to the Amazon rainforest[14]. There is growing evidence that mangroves have Total Ecosystem Carbon Stocks (TECS), which includes aboveground and belowground plant biomass, deadwood mass, and soil carbon pools, that are at least 2-fold those of upland evergreen forests[15]. For example, Brazilian mangroves have TECS between 413 and 1851 Mg C ha$^{-1}$ [7,16–18]; which are substantially above the C stocks of Brazilian upland forests such as the Amazon (463 Mg C ha$^{-1}$), Caatinga (74 Mg C ha$^{-1}$) and Cerrado (49 Mg C ha$^{-1}$)[14,19,20]. A global synthesis revealed that diverse environmental, climatic and physiographic gradients along coastal margins worldwide results in a broad range in the amount of carbon stored in mangrove forests (79–2208 Mg C ha$^{-1}$)[7].

Deforestation of mangroves is a global concern since their conversion to pastures, shrimp ponds or aquaculture permanently eliminate large C stocks from those forests. As a result, mangrove deforestation significantly contributes to GHG emissions. Unfortunately, field-based emission factors (EF) that allow accurate estimates of $CO_2$ release (or equivalent $CO_{2e}$) per area once mangroves are converted to other land uses are limited in many countries[11]. This limitation in field data drove attempts to model global mangrove carbon stocks and emission factors upon their loss, which are key to climate change policy[21,22]. However, the limited field assessments from regional datasets decrease the quality of such models, sometimes leading to large inaccuracies of the in-situ total ecosystem carbon stocks[7,23]. The same limitations exist in the regional availability of emission factors from mangrove conversion. Regional estimates from NE Brazil, the Caribbean, and Indonesia indicate that between 58 and 90% of the TECS are lost when mangrove forests are converted to shrimp aquaculture or cattle pastures[24–26]. As these emission factors are used to model GHG emissions to the atmosphere, precise assessments from field-based plots are critical to support regional mitigation programs. The large GHG emissions from Land Use and Land Cover Change (LULCC) in mangroves underscore their importance to inclusion in INDC and REDD+ strategies, and in Brazil's mitigation efforts within the UNFCCC.

Amazon mangroves form amongst the largest and most intact of coastal forests globally. To support their inclusion within the Amazon biome REDD+, it is necessary to establish deforestation reference levels, mangrove EFs related to LULCC activities and to expand their TECS assessment to the large coastal province where they are located, particularly near the Amazon River delta[27]. In this study, we provide a comprehensive regional accounting of mangrove TECS and EFs from LULCC in the Amazon coast based on 900 soil samples and tree measurements from over 190 forest plots (Fig. 1; Supplementary Table 1). We investigated the variability in mangrove TECS across the main hydrogeomorphic settings in the region and sampled mangrove-deforested areas (due to pasture and shrimp farms) in the Amazon to determine EFs of typical land use activities. We then estimated mangrove deforestation rates (i.e., activity data) in the Amazon coast from open-source satellite monitoring of forest cover to provide the first basis for their use into the Brazilian REDD+ program and calculated potential emission reductions in the best-case conservation scenarios. We find that Amazon mangroves hold total ecosystem carbon stocks that are below global averages, with annual deforestation rates of nearly 750 ha. Conversion of mangroves to pastures are a significant source of disturbance in the Amazon coast which needs to be mitigated and incorporated to the Brazilian iNDC and REDD+ program.

## Results

### Amazon mangrove ecosystem carbon stocks

The TECS of Amazon mangroves averages 468 ± 67 Mg C ha$^{-1}$ with a range between 181 and 903 Mg C ha$^{-1}$ (Supplementary Table 2). The aboveground mangrove biomass and total soil carbon ranges from 29 to 335 and 90 to 541 Mg C ha$^{-1}$, respectively. The TECS in estuarine and delta mangroves are similar (486 ± 21 and 478 ± 212 Mg C ha$^{-1}$, respectively). Mangroves occurring on open-coast hydrogeomorphic settings have 21% less carbon when compared to those in estuarine and delta settings (377 ± 160 Mg C ha$^{-1}$, PERMANOVA F = 30.72, $p$ = 0.01; Fig. 2; Supplementary Table 3). Total soil carbon is higher in estuarine mangroves (329.3 ± 19 Mg C ha$^{-1}$) when compared to delta and open coast settings (259 ± 124 and 207 ± 111 Mg C ha$^{-1}$, respectively; PERMANOVA F = 6.93, $p$ = 0.02; Supplementary Fig. 1). Organic carbon (SOC) of the surface 1-meter soils only accounts for 19–26% of the total soil C (300 cm) in Amazon mangroves. SOC density of the surface-1m soils differs across geomorphic settings, being higher on estuarine and delta mangroves (15.9 ± 1 g.cm$^{-3}$ and 16.8 ± 3.9 g.cm$^{-3}$, respectively) when compared to open coast settings (12.1 ± 3.4 g.cm$^{-3}$; PERMANOVA, F = 13.7; $p$ = 0.01; Fig. 3). The total aboveground carbon stocks are similar across the studied hydrogeomorphic settings and ranges from 129 ± 52 Mg C ha$^{-1}$ on open coast forests to 133 ± 12 Mg C ha$^{-1}$ and 181 ± 77 Mg C ha$^{-1}$ on estuarine and delta mangroves, respectively. Downed wood is higher in estuarine settings when compared to delta and open coast mangroves (PERMANOVA F = 4.21, $p$ = 0.03).

We additionally tested for the effects of climate (precipitation), hydrology (soil salinity), latitude, and tree basal area on the mangrove TECS in the Amazon. Among the studied sites, there are limited east-west gradients in precipitation (2000–2300 mm.yr$^{-1}$), tidal range (4–6 m), and latitude; therefore these variables are not significant to explain the regional variability in TECS. Although we find a significant relationship between tree basal area and mangrove TECS, the model explained about 30% of the variability in carbon stocks on the Amazon coast (F = 3.77, Adj $R^2$ = 0.3071; Supplementary Table 4). We used the Akaike Information Criterion (AIC) to select the best model explaining mangrove TECS on the Amazon coast, which improved the variation in TECS to 36% by using only tree basal area and soil salinity (F = 8.158, Adj $R^2$ = 0.364; Supplementary Table 5; Supplementary Fig. 2).

The organic matter isotopic signatures in mangrove soils are markedly distinct along the Amazon coast, indicating variable contribution from riverine or marine ecosystems (Supplementary Fig. 3; Supplementary Table 6). Mangrove soils and freshwater riverine tidal forests on the Amazon Delta receive a higher contribution of allochthonous freshwater sources than estuarine mangroves to the east of the Amazon River mouth which have a mixed contribution of freshwater and mangrove (autochthonous) plant sources. Marine organic sources (e.g., phytoplankton) have a minor contribution to the organic matter in soils of open coast mangroves but are typically present in shrimp ponds, indicating a higher contribution of allochthonous organic sources with land use change.

### LULCC effects in Amazon mangroves

Pristine (paired) estuarine mangroves near shrimp farms and pastures in the Amazon have TECS of 457 ± 82 Mg C ha$^{-1}$, whereas the converted sites (N = 4) average 123 ± 79 Mg C ha$^{-1}$ (PERMANOVA F = 19.59, $p$ = 0.01; Supplementary Table 7). We observe a complete loss of the aboveground stocks, and 71% to 83% decrease in TECS when mangroves are converted to shrimp ponds and pastures, respectively (Table 1). The TECS loss of mangrove-converted areas (shrimp ponds and pastures) in the Amazon average 351 Mg C ha$^{-1}$. The SOC density on the top-1m decreases from 20.1 ± 3.5 g.cm$^{-3}$ in estuarine mangroves to 5.5 ± 1.2 g.cm$^{-3}$ in converted shrimp ponds and pastures (PERMANOVA F = 21.96, $p$ = 0.04; Supplementary Fig. 4). Soils in converted sites have also higher average bulk density (1.7 ± 0.22 g.cm$^{-3}$) when compared to paired (pristine) mangrove forests (0.8 ± 0.23 g.cm$^{-3}$; PERMANOVA F = 19.29, $p$ = 0.02). The conversion of mangroves in the Amazon has a mean potential greenhouse gas emissions of 1228 ± 146 Mg $CO_{2e}$ ha$^{-1}$. The emissions from mangroves in the Legal Amazon is 3–20-fold higher than land use emissions in Brazilian upland biomes,

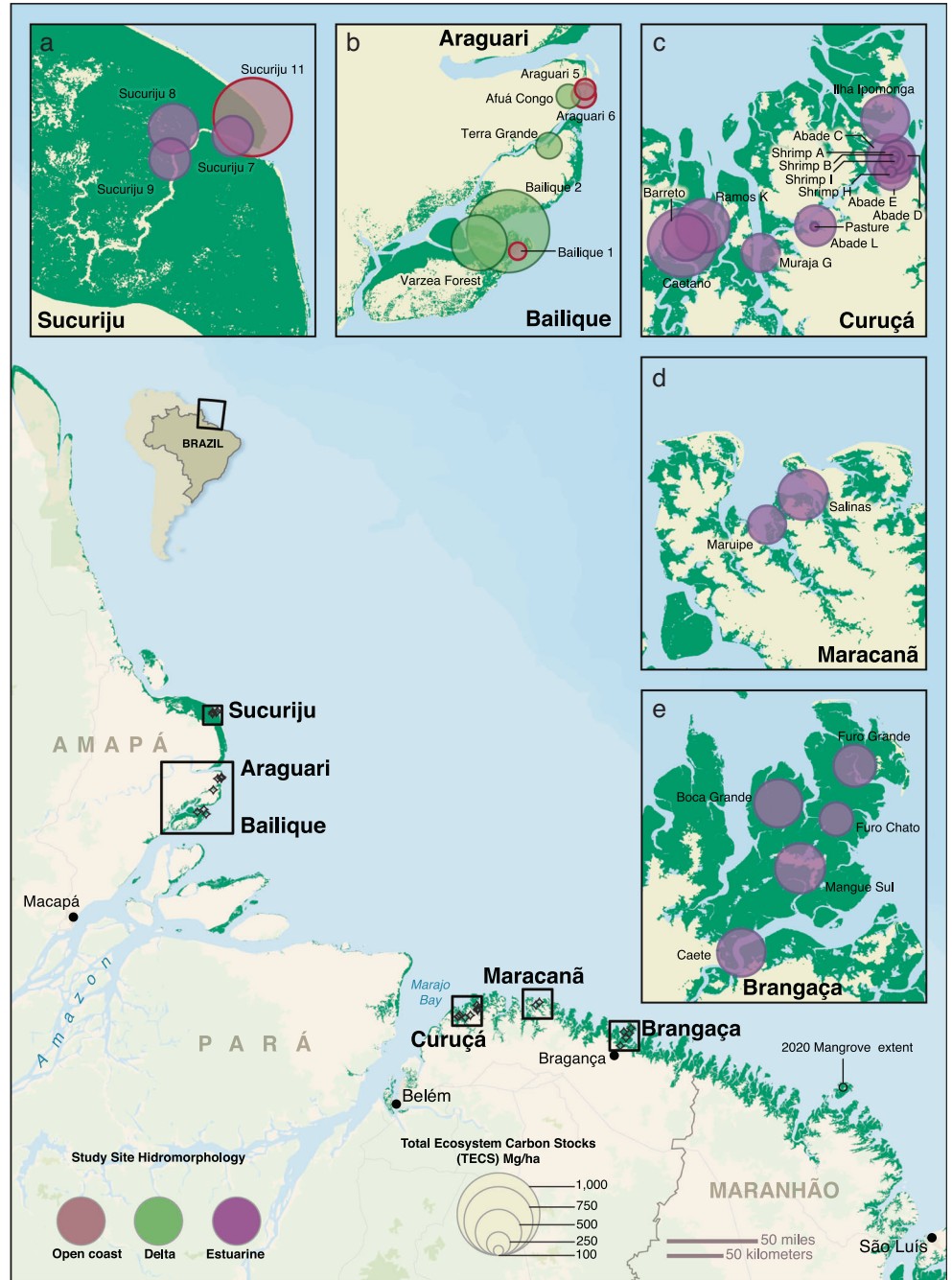

**Fig. 1 | Map of Amazon mangroves indicating the six main regions studied.** Areas to the north of the Amazon River mouth including Sucuriju (**a**), Araguari, and Bailique (**b**), and to the east including Curuçá (**c**), Maracanã (**d**), and Bragança (**e**). Shrimp farms (*n* = 3) and the pasture site (*n* = 1) were sampled in the Curuçá region (**c**). Color codes indicate hydrogeomorphic classifications (Open coast, Delta, Estuarine) and circle sizes indicate Total Ecosystem Carbon Stock (TECS) classes. Source data are provided as a Source Data file. Map created using ArcGIS® software by Esri Subscription ID 3968399452. ArcGIS® and ArcMap™ are the intellectual property of Esri and are used herein under license. Copyright © Esri. All rights reserved.

including the Amazon rainforest (331 Mg CO₂e ha⁻¹), the Cerrado and Caatinga biomes (55–131 Mg CO$_{2e}$ ha⁻¹; Table 2).

### Amazon mangrove reference emissions

We estimate from satellite land cover mapping that the Legal Amazon has 795,637 ha of mangrove forests. The mean annual mangrove deforestation in the Legal Amazon between 2016 and 2021 was 751 ± 248 ha (FREL; Supplementary Table 8). The rate of mangrove deforestation in the Amazon are limited (<0.1% of the total mangrove area), but over 94% of the annual deforested area is due to conversion to pastures or grasslands, suggesting that protection measures are necessary to avoid emissions through LULCC in the future. Based on the 2016–2021 mangrove FREL, we estimate annual emissions of 0.9 Tg CO$_{2e}$ ha⁻¹ from the conversion of mangroves to pastures and grasslands in the Amazon (Table 2). The Brazilian national FREL for its six upland biomes[5] (i.e., Amazon, Cerrado, Caatinga, Atlantic Forest, Pampa, and Pantanal) is set at 461 Tg CO$_{2e}$ ha⁻¹, based on a mean annual deforestation of 2 million ha yr⁻¹ (Table 2). The annual mangrove LULCC emissions correspond to 0.2% of Brazil's FREL from an area of less than 0.05% of the annual loss from upland biomes. Mangrove deforestation in the Amazon has decreased recently (2016–2021), which is an opposite trend of an increasing deforestation

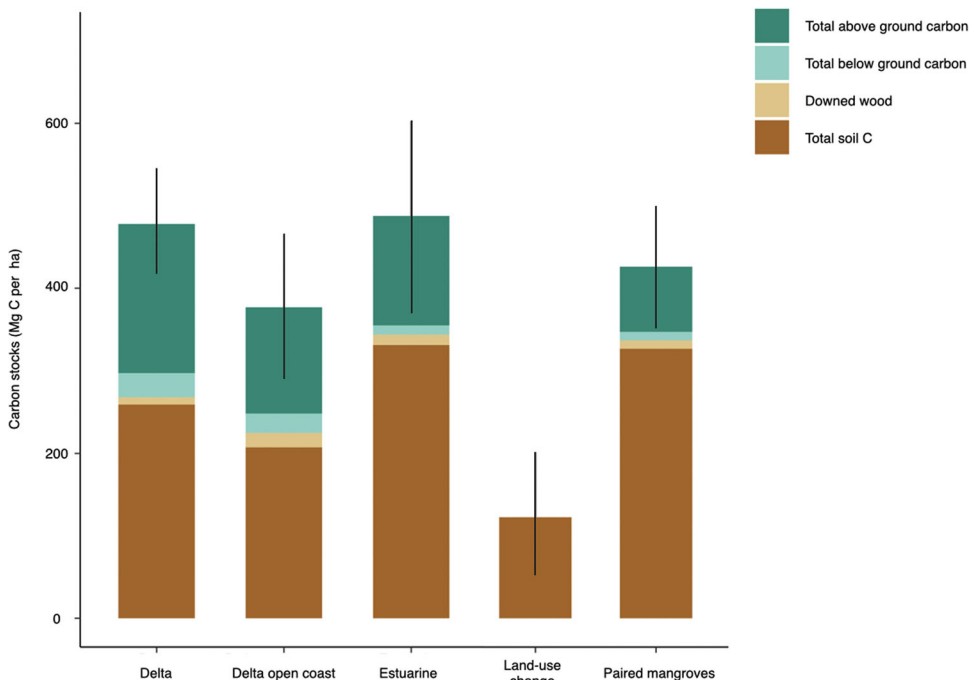

**Fig. 2 | The total ecosystem carbon stocks (TECS) of mangrove forests on the Amazon coast.** Bars represent studied hydrogeomorphic settings (Delta, delta open coast, and estuarine), and with land-use change (converted to shrimp ponds and pastures) and paired mangroves near converted areas. Error bars are 1 standard deviation. Source data are provided as a Source Data file.

from Brazilian upland biomes[5]. Considering a best-case scenario, actions to avoid mangrove loss from land-use change in the Legal Amazon could potentially avoid emissions of $9.2 \pm 0.11$ Mt $CO_2$e over a period of 10 years, which can be developed in projects used under the REDD+ framework, suggesting they are of great value to mitigate emissions from the forestry sector and finance biodiversity conservation.

## Discussion

Based on a comprehensive forest stock assessment, we find that Amazon mangroves hold total ecosystem carbon stocks of 468 Mg C ha$^{-1}$ (range of 180–902 Mg C ha$^{-1}$), which are 40% lower than the global average of 856 Mg C ha$^{-1}$ but within the range of mangrove forests globally (79–2208 Mg C ha$^{-1}$)[7]. Our dataset supports a strong role of soil depth as a key component of mangrove TECS, explaining the lower stocks of fringe/open coast mangroves which had shallower soil depths (<100 cm). Our data does not support differences in mangrove TECS among hydro-geomorphological settings including estuaries, delta and open cost mangroves in the Amazon coast[28,29]. The mean TECS of Amazon mangroves cannot be attributed to the proximity of sediment sources such as the Amazon River mouth, even though soil salinity and tree basal area partially correlated to the variability of mangrove TECS in the Amazon coast. The high regional variability in mangrove TECS within estuaries and other hydro-geomorphological settings suggest localized effects on carbon sequestration, which could result from the interplay of soil mineralogy, sea level rise, and coastal dynamics processes[29–32]. Organic carbon accumulation rates in estuarine mangroves to the east of the Amazon Delta are markedly lower (30–59 g m$^{-2}$ yr$^{-1}$ [33]) when compared to other Brazilian mangroves[17,18] and to the global average of 138 g m$^{-2}$ yr$^{-1}$ [34]. Carbon accumulation rates and the relative contribution of allochthonous sources of carbon are of additional interest in areas with large intact forests such as the Brazilian Amazon coast as preserved mangroves may contribute to climate mitigation by removing carbon from the atmosphere[9,17]. Further assessment of C accumulation rates in mangroves of the Amazon Delta will be key to understand the potential

effects of high riverine (allochthonous) mineral input on carbon stabilization at those sites as vertical accretion rates are positively correlated with sediment supply[30,33]. It is also possible that mechanisms that accelerate organic matter decomposition or mechanisms behind SOC interactions with soil minerals within Amazon soils may be a key factor to soil C stocks in the region[32,35,36].

Our dataset offers a critical step forward to refinement of global models of blue carbon, and highlight the unparallel value of field assessments to support carbon crediting offsets. For example, we found that SOC density was generally lower (range 8.6–22.9 mg C cm$^{-3}$; Table 3) than model estimates for mangrove forests along the Amazon coast[22,28]. Our dataset additionally indicates lower (<50%) soil organic carbon stocks in Amazon mangroves when compared to current global models for the Brazilian territory. As a result, our study supports an important refinement of global models using mangrove TECS losses from land use change in the Amazon region. The IPCC Tier 2 emission factors of conversion of mangroves to pastures and shrimp farms in the Legal Amazon were 71–78% of TECS, which is higher than previous emissions for the Brazilian territory (Table 3)[8,25]. Emissions arising from mangrove conversion are the highest of any terrestrial Brazilian biome and over 3-fold those from deforestation in the Amazon rainforest, which reveals the critical importance of protecting mangrove forests from deforestation.

Limited sampling within countries have commonly resulted in the omission of mangroves and other coastal wetlands for international INDC's and regional estimates (IPCC Tier 2) of GHG emissions arising from land use. In countries that hold extensive wetlands and potentially high C stocks, mangroves can have a far higher implication to national climate change frameworks[10,21,37]. In Indonesia, while mangroves correspond to 2% of the country's forest area, they account for 10% of total greenhouse gas (GHG) emissions from deforestation[21]. The conversion of mangroves to pastures in the Legal Amazon accounted for over 93% of the total area loss (Supplementary Data). Our dataset reveals an important pressure of agricultural land-use change in Amazon mangroves, which is also a major driver of loss to Amazon upland forests[14]. Based on the historical deforestation of mangroves in

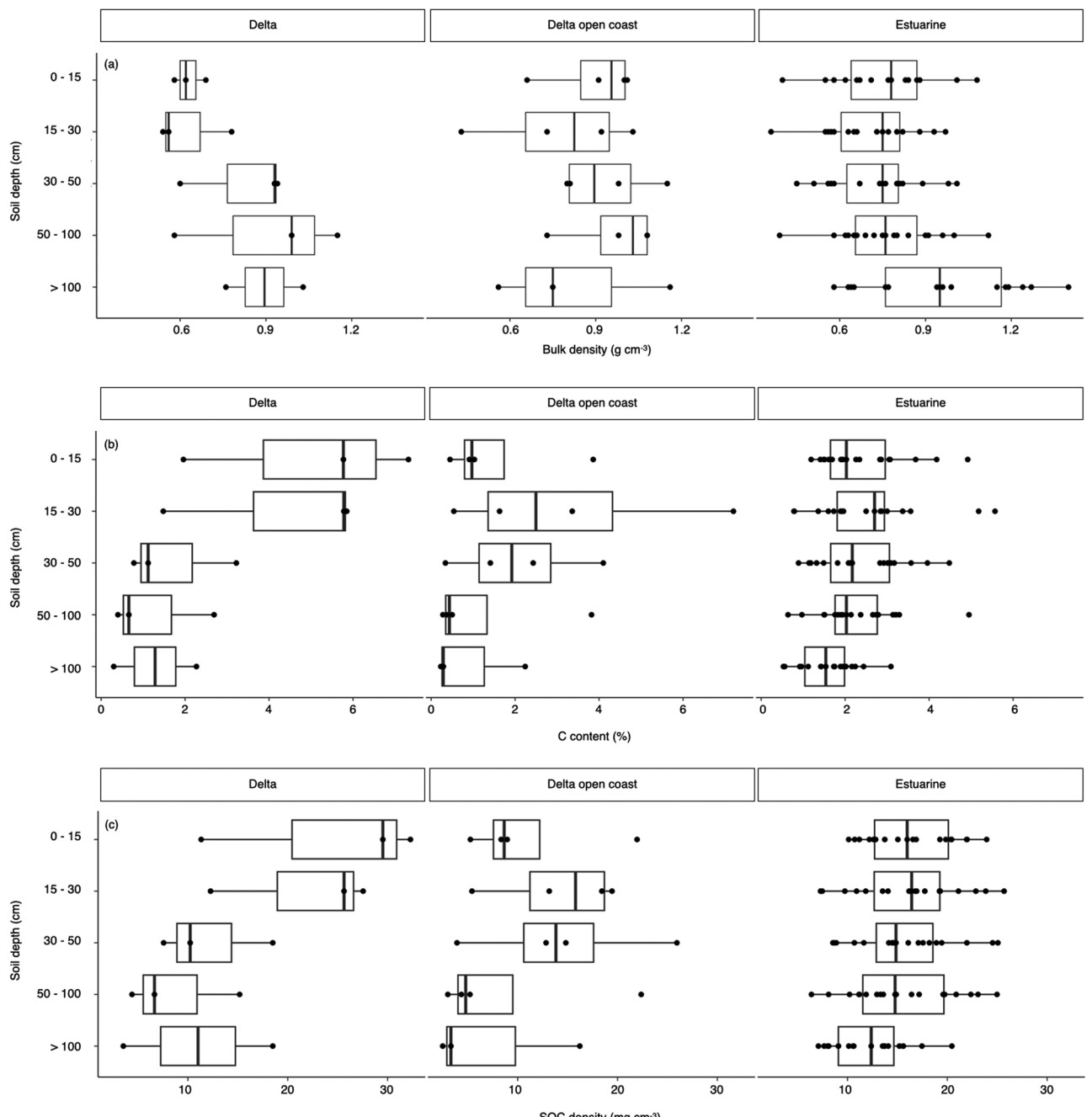

**Fig. 3 | Soil properties of mangrove forests according to hydrogeomorphic settings. a** Bulk density; **b** C (carbon) content and **c** SOC (Soil organic carbon) density per soil depth range. Boxplots bars represent the lower quartile, median and upper quartile, whiskers indicate minimum and maximum values, and dots indicate data points above or below 1.5× the quartiles. Source data are provided as a Source Data file.

the Amazon (0.12% annually) and total GHG emissions from LULCC, we could determine a regional Forest Reference Levels that could be submitted under the Paris Agreement. Assuming future Business as Usual conversion rates, the potential annual avoided emissions if mangrove conversion to pastures in the Legal Amazon were mitigated under a REDD+ program are $0.9 \pm 0.3\,\mathrm{Tg\,CO_2e\,yr^{-1}}$. This climate mitigation potential is nearly 50% higher than previously estimated by global models[38], and could potentially generate a significant market value to support an enhanced governance and protection of mangroves in the Legal Amazon. The low deforestation rates and high market value through conservation of mangroves in the Amazon are in contrast to all upland Brazilian biomes which are experiencing an increased or a continued rate of deforestation and GHG emissions

from land use change during the last decade[2]. Conserving mangroves in the Legal Amazon can offer additional natural net sinks and mitigation of GHG emissions through forest productivity, carbon burial, and co-benefits to coastal communities such as fisheries, coastal protections, wildlife habitat, and increasing local climate resilience[39].

Although some uncertainties exist with our upscaled results due to our study representing a limited number of land-use sites from estuarine mangroves, our estimates of TECS loss in the Amazon coast should be considered conservative. This is because the emission factor ratios from land use on the Amazon coast (73%) are within the range of global assessments (82%)[25], which for many areas adopted Tier 1 data[12] (IPCC standards) to estimate emissions given the difficulty in obtaining site-specific information (Table 3). In addition, our results are also

**Table 1 | Mean C stocks in Amazon mangroves and losses from deforestation**

| Land cover | Carbon pool | Mean | 95% CI | Propagated uncertainty |
|---|---|---|---|---|
| Amazon mangroves | AGB | 137.6 | 26.9 | |
| | BGB | 15.3 | 3.0 | |
| | Downed wood | 12.9 | 2.9 | |
| | Soil C | 303.8 | 49.4 | |
| | TECS | 468.3 | 67.4 | 32.9% |
| Shrimp pond | AGB | 0 | | |
| | BGB | 0 | | |
| | Downed wood | 0 | | |
| | Soil C | 133.8 | 84.8 | |
| | TECS | 133.8 | 84.8 | - |
| Pastures | AGB | 0 | | |
| | BGB | 0 | | |
| | Downed wood | 0 | | |
| | Soil C | 78.0 | 14.5 | |
| | TECS | 78.0 | 14.5 | - |
| Paired pristine mangroves | AGB | 82.4 | 25.5 | |
| | BGB | 10.0 | 1.5 | |
| | Downed wood | 10.6 | 4.1 | |
| | Soil C | 354.1 | 77.8 | |
| | TECS | 457.1 | 82.4 | 60.4% |
| Emission factors | | | | |
| | | TECS loss (per ha) | EF per ha (95% CI) | |
| Stock change | | | | |
| Shrimp ponds | | 323.3 | 1186.5 | |
| Pastures | | 379.2 | 1391.5 | |
| Mean LULCC Amazon mangroves | | | 1228 (146) | |

Reported carbon pools of pristine and converted mangrove forests with their confidence intervals (CI) and Propagated uncertainty. Emission factors (EF) correspond to gross emissions (Mg CO₂e), based on the stock change approach compared to paired pristine mangroves in the sub-region where pastures and shrimp farms were sampled. All stock values are given in Mg C ha⁻¹.
*AGB* above ground biomass, *BGB* below ground biomass, *TECS* total ecosystem carbon stocks, *LULCC* land use land cover change.

within the range of emissions from land use in upland dry and evergreen forests, which results in 29-80% of carbon losses after pasture establishment[40]. Increasing field assessments in mangrove-converted areas to pastures and agricultural lands would be an important step to incorporate natural and human effects on these estimates. The use of forest carbon offsets globally have received some criticism concerning double-accounting of credits between voluntary carbon markets and INDC's, and with many doubts concerning recent REDD+ project's effectiveness[41]. Problems with double-accounting and better practices to monitor REDD+ project's success will need to be addressed by the UNFCCC and in international carbon markets.

Similar to mangroves throughout the world, protecting Amazon mangroves would be far more efficient in offsetting GHG emissions, in terms of per unit area, when compared to terrestrial forests, savannas, and grasslands of Brazil. For example, the potential annual mangrove mitigation in the Legal Amazon (751 ha yr⁻¹) is equivalent to the annual net carbon accumulation in 82,400 ha of secondary forests[42]. Mangrove reforestation in Brazil where deforestation and conversion to aquaculture is quite significant could offer additional mitigation benefits[24,43]. Although the mangrove mitigation scenarios are highly optimistic considering the challenges to control deforestation and difficulties in setting forest reference levels[4], they support an IPCC Tier 2 assessment for an extensive mangrove area in the Legal Amazon with detailed regional C stocks and activity data. In this way, Amazon mangroves may offer a similar mitigation potential to other preserved wetlands or in areas such as SE Asia where underpinning restoration programs are being implemented[37,43].

## Methods
### Study sites and field sampling
The Amazon River delta receives the largest global freshwater and sediment discharge of any coastal oceans, which may create wide environmental variability to the flux dynamics and consequently in mangrove TECS and organic sources over time, which has not been assessed to this day. Considering this broad regional variability, we attempted to sampled mangroves across estuarine (or riverine), delta, and delta/open coast hydrogeomorphic settings within the Brazilian Legal Amazon (Supplementary Table 1; Fig. 1). Within each site, ecosystem carbon stocks (aboveground and belowground) were measured following standard methods outlined by Kauffman and Donato[44]. At each mangrove and shrimp pond or pasture site, six plots were established 20 m apart along a 100 m transect positioned in a perpendicular direction from the mangrove/estuary or coast ecotone. Due to the limited accessibility to private land, we sampled four shrimp ponds and one pasture site within the Legal Amazon. Control undisturbed mangrove plots were sampled next to these sites in order to remove the regional variability in stocks from LULCC emission factors

**Table 2 | National Forest Reference Emission Levels (FRELs) of Brazilian upland biomes**

| | Total area (ha) | Ecosystem stocks (Mg C ha⁻¹)** | Mean deforestation (ha yr⁻¹) | Emission Factor (Mg CO₂e per ha) | Annual emissions (Tg CO₂e yr⁻¹ ± 95% CI) | Biome emissions relative to national FREL |
|---|---|---|---|---|---|---|
| Legal Amazon mangroves | 795,637 | 468 (181–903) | 751 | 1228 | 0.9 ± 0.3 | 0.2% |
| Amazon biome* | 421,274,200 | 129 (34–604) | 1,055,529 | 331 | 349 ± 116 | 75.7% |
| Cerrado biome | 198,301,700 | 41 (18–131) | 570,222 | 55 | 31.5 ± 8.6 | 6.8% |
| Caatinga biome | 86,281,800 | 41 (18–186) | 200,867 | 131 | 26.3 ± 1.8 | 5.7% |
| Atlantic forest biome | 110,741,900 | 123 (15–183) | 84,807 | 409 | 34.7 ± 7.5 | 7.5% |
| Pampa biome | 19,381,800 | 9 (5–174) | 39,971 | 102 | 4.1 ± 0.9 | 0.9% |
| Pantanal biome | 15,098,800 | 72 (12–168) | 26,453 | 228 | 6.0 ± 1.1 | 1.3% |

Upland biomes data submitted to the UNFCCC in the context of results-based payments for reducing emissions (REDD+)[5]. Brazil's National FREL for the land use sector is set at 461 Mt CO₂e.
*Amazon biome area that does not include mangroves.
**Brazil's submission to the UNFCCC do not account for soil C stocks on the upland biomes.

**Table 3 | Comparison of Legal Amazon mangrove data from this study with previous estimates from the literature for the Brazilian Northern coast**

| Reference | N | Soil C stocks (1 m) | Total soil C | SOC density 1 m | TECS | Soil C stocks loss per ha | TECS loss per ha | SOC EF (Shrimp ponds/ pasture) |
|---|---|---|---|---|---|---|---|---|
| Legal Amazon mangroves (this study) | 780 | 146 (20) | 302 (49) | 15.4 (2) | 468 | 248 | 334 (91) | 0.62/0.78 |
| Atwood et al. (ref. 21) | 41 | 308 (130) | | | | | | |
| Rovai et al. (ref. 27) | 36 | | | 30.9 (5) | | | | |
| Sanderman et al. (ref. 22) | <50 | 340 (76) | | | 330 | | | |
| Kauffman et al. (ref. 7) | 90 | | | | 647 | | | |
| Adame et al. (ref. 12) | | 363 | 724 | | 473 | | | 0.67/0.33 |

SOC density (integrated to 1 m soils), and associated losses after disturbances in Brazil where available. TECS loss per ha. Stock losses from land use disturbances based on paired undisturbed mangroves. SOC EF. Reported emission factor rates for soil organic carbon. All other data integrated to 3 m. Stock data in Mg C ha$^{-1}$, SOC density in mg cm$^{-3}$.
*TECS* reported values of total ecosystem carbon stocks, *SOC* soil organic carbon, *N* Number of reported soil samples collected or used to model data for Brazilian mangroves.

(see below). At each plot, we collected data necessary to calculate total carbon stocks derived from standing tree biomass, downed wood (deadwood on the forest floor), and soils to the depth of an indurated horizon composed of marine sands.

## Biomass of trees and shrubs

The typical species of mangroves for the Brazilian biogeographic province, including *Rhizophora* mangle L. (Rhizophoraceae), *Avicennia germinans* (L.) Stearn (Avicenniaceae), *Laguncularia racemosa* (L.) Gaertn., (Combretaceae) were present at sampled sites. However, a unique feature of mangroves in the Amazon delta was a mix of freshwater and *varzea* plants, including Buriti and Acaí palms (*Euterpe oleracea*) and Cortiça trees (*Pterocarpus* sp.)[27]. In each plot we recorded the composition, tree density, and basal area of the mangroves and other plants. Tree diameter was obtained at 1.3 m height (diameter at breast height, dbh) of all trees within each plot of each transect within an area of 154 m² (7 m radius) for trees over 5 cm dbh, and a 2 m-radius nested plot for trees with a dbh of less than 5 cm. The diameter R. *mangle* was measured at 30 cm above the main branch on the highest prop root.

The biomass of trees was determined using allometric equations based on preliminary work developed for mangrove species encountered in this study. The equations utilized were species-specific, from similar environmental conditions, and included the range in tree diameters measured in our plots. The allometric equations used represent a range of mangrove species and sizes found in this study[45,46]. We also used allometric equations to derive tree mass for the varzea trees measured in the Amazon delta[47,48].

Mangrove belowground root biomass was calculated based on equations developed by Komiyama, et al.[49]. The tree carbon content (C) was estimated by multiplying 0.48 and 0.39 for aboveground and belowground biomass, respectively[44]. We included the standing dead trees into the aboveground biomass calculations, where their dbh was measured and given one of three decay classes: 1 (dead trees without leaves), 2 (dead trees without secondary branches), and 3 (dead trees without primary or secondary branches). Biomass of class I dead trees was estimated to be 97.5% of a live tree, class II—80% of a live tree, and class III—50% of a live tree[44].

## Downed wood

The mass of dead and downed wood was calculated by the planar intersect technique[44]. Four 14-m transects were sampled starting at the center of each plot in a direction that was offset 45° from the azimuth of the main transect. The other three transects followed a 90° clockwise from the previous transect, and in each transect, the diameter of any downed wood intersecting the transect was measured. Downed wood equal o higher than 2.5 cm but with less than 7.5 cm in diameter at the point of intersection were measured along the last five meters of

the transect. Downed wood ≥7.5 cm in diameter at the point of intersection were measured from the second meter to the end of the transect. Downed wood was classified as sound or rotten, where the latter was used for wood that were visually decomposed and fragile when impacted. The wood mass was estimated based on the specific gravity of mangrove downed wood[50]; and converted to C using a factor of 0.50[44].

## Soil carbon

The soil carbon was sampled at the center of each plot from a fixed-volume soil sample from a peat auger consisting of an open-faced cylindrical chamber with a 6.4 cm radius. Soils were also used to determine soil bulk density. The core was divided into depth intervals of 0–15 cm, 15–30 cm, 30–50 cm, 50–100 cm, and >100 cm (if parent materials or an indurated horizon were not encountered before 100 cm depth). We determined the maximum soil depth in all plots using a graduated aluminum probe. The soil carbon pools were calculated to 3 meters when max soil depth was over this limit. All samples collected in the field were dried at 60 °C to a constant mass and then weighed to determine bulk density. Laboratory analysis was conducted at the Southern Cross University Biogeochemistry Lab, Australia. Soil concentration was determined using a Thermo Flash EA 1112 series C-N Soil Analyzer. A total of 914 soil samples were collected in this study and analyzed for total carbon. Soil salinity (range 0 to 100) and pH were measured from a handheld refractometer and pH meter (ATC−QC-PH-02), respectively. We strived to measure the borehole salinity and to avoid mixing with surface water that was usually lower in salinity due to rainfall. We measured salinity and pH sampled at each soil sampling plot (n = 6 in each sampled stand).

## Stable isotopes

Soil samples were dried (0.5 mg) and acidified (1% HCl) in tin or silver boats to remove inorganic carbon. Samples were combusted in a Eurovector elemental analyzer and resulting $N_2$ and $CO_2$ gases were separated by gas chromatography and admitted into an IRMS mass spectrometer for determination of $^{15}N/^{14}N$ and $^{13}C/^{12}C$ ratios (reproducibility: 70.5% for d$^{15}$N and 70.2% for d$^{13}$C). C-isotopic ratios were measured against a Pee Dee Belemnite (PDB) standard for $\delta^{13}$C and atmospheric nitrogen for $\delta^{15}$N. Results are expressed as delta ($\delta$) notation, where $\delta X$ (‰) = [(Rsample/Rstandard) − 1] × 103, where R = $^{15}N/^{14}N$ or $R = ^{13}C/^{12}C$.

## Amazon mangrove LULCC emission rates

Greenhouse gas emissions from deforestation were estimated based on the difference from the total ecosystem carbon stocks (aboveground, belowground, deadwood, and soils) of shrimp farms and pastures from paired mangroves in the Amazon (IPCC Tier 2). The emission factor (EF) from mangroves accounts for the entire soil

profile or a default depth of 3 m when soils exceeded this depth. We calculated the propagated uncertainty in the estimation of total ecosystem carbon stocks considering the confidence interval of each stock estimate (AGB, BGB, Downed wood, and Soil C) following the GOFC-GOLD methods[51]. Annual emissions were obtained by the product of the mean annual deforested area by the average area-based emission factors (EF) from conversion of mangroves to pastures in the Amazon. The ecosystem losses are reported as potential $CO_2$ emissions, or $CO_2$ equivalents ($CO_2$e) obtained by multiplying C values by 3.67, the molecular ratio of $CO_2$ to C. While reported as the $CO_2$e, these estimates account only for changes in ecosystem C in situ.

### Amazon mangrove mitigation potential

The inclusion of Amazon mangroves in Brazil's REDD+ carbon markets require activity data on historical mangrove deforestation and emission factors from land use impacts. Activity data was obtained from the MapBiomas annual Land Use and Land Cover Mapping project[52]. This project provides annual estimates of deforestation and land use and associated classifications (e.g., pastures, grasslands), which were used to derive mean mangrove loss rates in the Amazon biome. We considered all losses of mangroves to pastures, grasslands, salt flats, and savannas as human-derived, as the removal of mangroves is typically the dominant driver of deforestation[23]. Over 93% of the mangrove deforestation in the Legal Amazon resulted in pastures establishment. Although the Paris Agreement sets the year of 2013 as a baseline to assess future emissions[53], we based the mangrove deforestation reference level in the Amazon as the most recent average annual deforestation between 2016 and 2021.

In 2023, Brazil submitted its National Forest Reference Emission Level (FREL), for the biomes of Amazon, Cerrado, Caatinga, Atlantic Forest, Pampa, and Pantanal[5]. These FRELs were used to determine the emission targets for the forestry sector and intended to be used in the REDD+ framework for payment based on mitigation of deforestation to the secretariat of the United Nations Framework Convention on Climate Change (UNFCCC). The carbon pools in Brazil's FREL only includes the aboveground and belowground biomass and carbon in litter and deadwood. As we attempted to compare the value of mangroves to that submission, we considered the carbon stocks reported to the UNFCCC. Carbon stocks, activity data (deforestation rates), and associated emissions from Brazil's upland biomes were obtained from the Government report submitted to the UNFCC[5].

### Statistical analysis

Differences between carbon stocks in mangroves across geomorphic settings and emissions from shrimp ponds and pastures (grouped as "impacted") were tested with a permutation analysis of variance (PERMANOVA[54]). If the PERMANOVA was significant, a least significant differences test was performed to determine which means were significantly different. Regional and local variability in mangrove carbon stocks (Total soil C, SOC density, downed wood C, TAGC, TBGC, TECS), forest characteristics (tree basal area and density), soil properties (soil salinity and bulk density), and soil $\delta^{13}$C signatures were evaluated by univariate PERMANOVAs (Euclidean distance), testing differences according to hydrogeomorphic settings (estuarine, delta, delta/open coast) and land-use change (pasture, shrimp pond). An $\alpha = 0.05$ was considered for all the analyses, datasets were log-transformed ($\log(x + 10)$ or $\log(x + 100)$) when required to allow multiple comparisons[55], and the differences within each group were identified by post hoc pairwise tests.

Multiple linear regressions between climatic (precipitation), hydrological (soil salinity, tidal range), forest structure (tree basal area) and geographic location (latitude, longitude) were fit to assess their degree of influence on mangrove TECS on the Amazon coast. After testing for multicollinearity among variables, we removed longitude

and tidal range from the regressions as they both have a strong east-west gradient (Pearson rank correlation > 0.5). Normality tests were run on model's residuals through QQ-plots and Shapiro–Wilk normality tests. After obtaining the multiple linear regression values, we used the Akaike Information Criterion (AIC[56]), through a stepwise backward model configuration. Graphical and analytical processing were performed in R project environment[57], using packages 'stats','vegan'[58] and MASS[59] for statistics and 'ggplot2' for charts[60].

## Data availability

Raw data have been deposited in Figshare and accessible at https://doi.org/10.6084/m9.figshare.23634579. Source data are provided with this paper.

## Code availability

R code are provided in the Source data with this paper.

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

## Acknowledgements

We thank many students and local villagers for their support in fieldwork, Martin Gamache for help with maps, and the Brazilian Ministry of Environment ICMBio for research permits. This work was supported by the National Geographic Society and Rolex Perpetual Planet Amazon Expedition (PFA-21-PPO31) grant to AFB. AFB and TOF are supported by the Brazilian National Council for Scientific and Technological Development (CNPq) through PQ fellowships. RFC is supported by a Brazilian Ministry of Education PhD fellowship (CAPES N 88887.481406/2020-00).

## Author contributions

AFB, TOF, and JBK contributed to study design, funding acquisition, sampling, analysis, and interpretation of data and wrote the manuscript; ACAM, RFC., FS, MO., GNN, and CJS contributed to sampling, analysis, and interpretation of data. AFB, ACAM, RFC, FS, MO, GNN, CJS, TOF, and JBK have approved the submitted version.

## Competing interests

The authors declare no competing interests.
