## [Peer review file · Nature Communications]

REVIEWER COMMENTS

Reviewer #2 (Remarks to the Author):

The study has collected and analysed substantial carbon data for above-ground biomass and soil from natural mangroves across Brazil's Legal Amazon and using established allometric equations and ratios to estimate carbon in above-ground and below-ground biomass, and at least 1m depth from soil cores. The work contributes significantly to our knowledge on the amount of mangrove ecosystem carbon stocks in Brazil and the risk of land-use change to their climate change mitigation potential.

The survey methods are sound, but I question why 100 m transects were established perpendicular to the coast, rather than parallel. For the natural mangroves, the number of survey sites is very comprehensive and across a range of geomorphic settings. However, the carbon stock data from the land use changes (ponds and pasture) is limited, from only 4 and 1 site, respectively. This is a major limitation of the study, as the avoided emissions were calculated based on the difference in carbon stocks between the paired sites (impacted and adjacent natural) at ponds and pasture. Given, over 93% of the mangrove deforestation in the Legal Amazon resulted in pastures establishment, basing the calculation off 1 field site for pasture is a limitation, and doesn't account for any natural or human variation in the pasture settings.

Furthermore, the mangrove FREL was based on the deforestation between 2000-2013, and although this is recommended by the Paris Agreement, this is an overestimate as annual mangrove deforestation rates have decreased since 2013. This should be changed to the more recent baseline (2016-2021) or the limitation acknowledged.

Also I acknowledge that the motivation for this study was probably to include Brazil's mangroves for results based payments under REDD+. However, I think the significance of this study is the significant mangrove forest carbon stocks, and the large losses to carbon stocks that agricultural land use change can have, and therefore that the focus of the paper should be on that, and then the potential benefits of including then in REDD+ included as a discussion point (i.e. a lot of the introduction can be moved to the discussion).

The work would also have greater significance to the field, if you are able to identify the environmental factors influencing biomass and soil carbon stocks. You have looked at the effect of geomorphic setting, but you could also investigate the effect of other factors, including ecological (e.g. tree and shrub species diversity), environmental (e.g. soils, hydrology, salinity) and climatic factors. Cite the body of literature on this for hypotheses on drivers of carbon accumulation.

Some detailed comments below.

Abstract – UNFCCC in full first.

47-51 Contradictory statements: For example, "Brazilian mangroves have TECS between 413 to 1851 Mg C ha⁻¹... However, diverse environmental, climatic and physiographic gradients along 50 coastal margins results in a broad range in the amount of carbon stored in mangrove forests (79 - 2,208 Mg C ha⁻¹)". Is the latter statement from literature? If so make this clear that it is.

72 – LULCC in full first

72 – 73 – explanation of how NDC, REDD+ and voluntary carbon market strategies all co-exist/fit together, and mitigation efforts within the United Nations Framework Convention on Climate Change (UNFCCC). I think it's important to note that there is potential for double counting if REDD+ used for voluntary carbon markets and contribution to country's NDC.

The paper is written like policy driving research, but I prefer the other way around – state importance first (contribution of mangroves to national carbon stocks), then place in context of policy needs to preserve the carbon stocks.

State in the introduction, what carbon stocks were included in the TECS -above-ground biomass,

below-ground biomass, downed wood, and soil carbon?

107 – 21% less carbon than what?

Fig 2 – need to spell TAGC and TBGC out in the caption. Can't see the other half of the error bar.

107-126 – no mention of the other carbon stocks and variation between hydrogeomorphic settings, but looks like downed wood is greatest in delta open coast, and BGC is proportional to ABC.

147 – The mangroves of concern are unclear in this sentence: "Mangrove TECS in the Amazon averaged 351.2 Mg C ha⁻¹ as a result of their conversion to shrimp farms and pastures."

174-176 – Is this estimate of avoided emissions based on the 2008-2013 or the 2016-2021 deforestation rates. Seems it would be an overestimate if based on the earlier. FREL for mangrove should be based on the later deforestation rate, given deforestation rates have declined.

175-176 – it is not the mangrove loss that would generate carbon credits, but avoided loss of mangroves from land-use change. And please clarify in the introduction how carbon credits are sold under the REDD+ framework.

190 – While the forest stock assessment is very comprehensive, I wouldn't agree that the land use change assessment is that comprehensive (4 shrimp ponds and 1 pasture). For ponds is acceptable, but wouldn't capture any variation for the pasture.

199 -201 Inclusion of other environmental variables in the analysis would make this assessment more robust and would confirm the influence of other environmental, climatic and physiographic factors, as you say these were important predictors of the amount of carbon stored in mangrove forests other than hydro-geomorphological settings in the introduction. For example, you could include species composition, salinity, bulk density, sedimentation/erosion, rainfall, temperature, . The role of other environmental variables is only briefly discussed and could be enhanced with citing other studies.

239 – 240 – You mean efficient in terms of per unit area. However, terrestrial ecosystems cover a greater area, so they may still offer similar avoided emissions, if deforestation rates are similar.

241 – State what the potential annual mangrove mitigation is – avoided clearing in XXXX ha of mangroves. And is this equivalent to the carbon accumulation in 300,000 ha of secondary forests, also annually?

Mention the stable isotope analysis in the results, but don't elaborate in the discussion, whether the higher soil carbon stocks in estuary mangroves has something to do with the source of carbon, e.g. because the estuaries have both freshwater allochthonous carbon and mangrove autochthonous carbon, whereas the deltas have mainly allochthonous?

Methods – what is a "delta/open coast" setting?

Only assessed land use effects in estuary mangroves, which have higher soil carbon than the other geomorphic settings, so applying the estuary emission factor to the potential loss of all mangroves across the Amazon is likely to be an overestimate, because some of this deforestation would be in delta and open coast mangroves, where the EF's are likely to be lower, because of lower initial soil carbon stocks. How do the EFs developed compared to other studies, e.g. Sasmito et al. 2019 <https://doi.org/10.1111/gcb.14774>

At each site, six plots were established, which is great replication. But why were plots established along a 100 m transect in a perpendicular direction from the mangrove/estuary or coast? That would capture zonation across the transect. Wouldn't it be better to establish parallel, then place the six plots to represent the different zones of mangroves, associated with species and structure? Please explain why you did the field design this way and any implications to the results.

Reviewer #3 (Remarks to the Author):

The manuscript titled "The inclusion of Amazon mangroves in Brazil's REDD+ program" is a good quality paper which highlighted how can we convert the carbon stock data from mangroves to REDD+ which is globally significant. The research was done in an important mangrove ecosystem of the world and used 196 plot data. This is very appreciating. The authors had done a tremendous field work to obtain the data. However, the manuscript can be accepted only after the following clarifications.

1. In many areas it is written that carbon stock can be used for mitigating climate change. The total ecosystem carbon stock assessment actually depicts vulnerability potential to climate change in the

case of its deforestation. The carbon accumulation rate per year actually will estimate its mitigation potential to climate change. Even a high total ecosystem stocked areas may be having low carbon accumulation rate due to LULCC especially when it is converted to aquaculture farm. Therefore, TECS assessment not alone can't be used for climate change mitigation. However, will be a good scientific document for REDD+ funding. The authors can rewrite and can emphasis on the importance of carbon accumulation rate assessment also for the accurate carbon credits. Refer : "Relevance and magnitude of 'Blue Carbon' storage in mangrove sediments: Carbon accumulation rates vs. stocks, sources vs. sinks". <https://doi.org/10.1016/j.ecss.2020.107027>

2. Also please confirm whether Tier 3 method is for carbon stock change over a period of time or just for carbon stock. The description of the methodology is not clear whether they used time series data or not?

3. Also refer IPCC 2019 refinement to 2006 guidelines if any changes are there for the methodology.

Minor Corrections

1. In Fig.2 mark the label as Land use change instead of land use also paired or pristine mangrove. Please define the classification properly.

2. Line no 154 The carbon emissions from mangroves in the Legal Amazon is 3-20-fold higher than land use emissions

3. Line no.153 CO₂ is not the single GHG gas. CH₄ emission is also very potential emission due to this aquaculture conversion. Therefore, rewrite the sentence.

Reviewer 2

The study has collected and analyzed substantial carbon data for above-ground biomass and soil from natural mangroves across Brazil's Legal Amazon and using established allometric equations and ratios to estimate carbon in above-ground and below-ground biomass, and at least 1m depth from soil cores. The work contributes significantly to our knowledge on the amount of mangrove ecosystem carbon stocks in Brazil and the risk of land-use change to their climate change mitigation potential.

We thank the reviewer for this comment

The survey methods are sound, but I question why 100 m transects were established perpendicular to the coast, rather than parallel.

Thank you. (Lines 317-326) We placed the transects perpendicular to the marine-mangrove ecotone, following the stock assessment protocol by Kauffman and Donato, 2012 (Kauffman, J.B. and Donato, D.C. 2012 Protocols for the measurement, monitoring and reporting of structure, biomass and carbon stocks in mangrove forests. Working Paper 86. CIFOR, Bogor, Indonesia). The advantage of this choice is that it provides a good estimate of changes in the composition of trees, structure, elevation and tidal gradients. This protocol had been widely used throughout the world for the assessment of C stocks in mangrove ecosystems. In addition, a parallel transect would not cover the range in environmental conditions encountered in mangroves as one moves inland from the marine/mangrove interface.

For the natural mangroves, the number of survey sites is very comprehensive and across a range of geomorphic settings. However, the carbon stock data from the land use changes (ponds and pasture) is limited, from only 4 and 1 site, respectively. This is a major limitation of the study, as the avoided emissions were calculated based on the difference in carbon stocks between the paired sites (impacted and adjacent natural) at ponds and pasture. Given, over 93% of the mangrove deforestation in the Legal Amazon resulted in pastures establishment, basing the calculation off 1 field site for pasture is a limitation, and doesn't

Thank you. The reviewer is correct in that we have limited replicates for the pasture conversion sites. Unfortunately, and as indicated in the methods (Lines 321-322), our access to private land where mangroves were replaced by pastures was limited.

However, we strived to overcome this by adding a paragraph in the discussion (Lines 281-294) highlighting the potential uncertainties from this limitation. We compared the emission factors from our sites (3 shrimp ponds and one pasture) with Sasmito et al., 2019 work and others,

account for any natural or human variation in the pasture settings.	and moved a former supplementary table to the main text for clarity. We highlight that our estimates are well within the range of global estimates for C loss in land-use settings (pastures, shrimp ponds and others, as highlighted by the global review of Sasmito et al., 2019), and are comparable to upland emissions from pasture conversions. We also found that pastures had similar carbon stocks as well as GHG emissions from pastures formed in Mexican mangroves of the Atlantic (Kauffman et al., 2016). As noted here, the emission factors found in this work, which were used to estimate potential emissions from mangrove loss in the Amazon are likely within the range of the natural and human variability, according to the current literature. However, we indicate more data is needed to reduce the uncertainties in these values.
Furthermore, the mangrove FREL was based on the deforestation between 2000-2013, and although this is recommended by the Paris Agreement, this is an overestimate as annual mangrove deforestation rates have decreased since 2013. This should be changed to the more recent baseline (2016-2021) or the limitation acknowledged.	Thank you. We corrected the avoided emissions estimate based on a FREL from 2016-2021 (Supplementary Table 8) and Lines 413-423
Also I acknowledge that the motivation for this study was probably to include Brazil's mangroves for results based payments under REDD+. However, I think the significance of this study is the significant mangrove forest carbon stocks, and the large losses to carbon stocks that agricultural land use change can have, and therefore that the focus of the paper should be on that, and then the potential benefits of including then in REDD+ included as a discussion point (i.e. a lot of the introduction can be moved to the discussion).	We do agree that the data provided on carbon stocks and losses due to land use is a great contribution of the paper to science and data are clearly presented for those with this interest. We also agree that the dataset offers a great opportunity to report regional analysis on drivers of C stocks and demonstrates the high emissions arising from land use change in the Amazon. To that end, we have included further analysis (as requested below) testing the importance of climatic and other variables to mangrove TECS, and discussed briefly the role of agriculture in emissions from mangrove conversion in Brazil. (Lines 265-267 and 295-299) However, we also believe a central focus on the role of mangroves in Brazil's

	REDD+ strategy will have a higher impact towards their conservation in this globally important region. This will be of great interest to policy specialists and decision makers in Brazil and throughout the tropical world. We moved parts of the introduction to the discussion as suggested by the reviewer (Lines 258-267)
The work would also have greater significance to the field, if you are able to identify the environmental factors influencing biomass and soil carbon stocks. You have looked at the effect of geomorphic setting, but you could also investigate the effect of other factors, including ecological (e.g. tree and shrub species diversity), environmental (e.g. soils, hydrology, salinity) and climatic factors. Cite the body of literature on this for hypotheses on drivers of carbon accumulation.	Thank you. We agree and now report multiple linear regressions to test for the association of climatic (precipitation), hydrological (salinity), latitude and tree basal area with total ecosystem carbon stocks (Lines 127-136; Supplementary tables 4 and 5). We included these variables as they could be easily obtained by weather stations, simple field measurements or satellite observations (Lines 448-457). After removing significant co-variables, we show that tree basal area had a significant, but limited power to predict TECS (Adjusted R²= 0.30). We then used the Akaike Information Criterion (AIC) to select the best model to predict TECS in the Amazon coast, and obtained a reduced model with soil salinity and tree basal area with a slightly better Adj R²=0.36. We now detail these results and discuss these patterns relating to the literature as the reviewer suggested (Lines 207-231).
Detailed comments	
Abstract – UNFCCC in full first.	Thank you. Corrected - Lines 24-25
47-51 Contradictory statements: For example, “Brazilian mangroves have TECS between 413 to 1851 Mg C ha⁻¹... However, diverse environmental, climatic and physiographic gradients along 50 coastal margins results in a broad range in the amount of carbon stored in mangrove forests (79 - 2,208 Mg C ha⁻¹)”. Is the latter statement from literature? If so make this clear that it is.	Thank you. This sentence was corrected for clarity (Lines 63-71)
72 – LULCC in full first	Thank you, corrected, Line 44

72 – 73 – explanation of how NDC, REDD+ and voluntary carbon market strategies all co-exist/fit together, and mitigation efforts within the United Nations Framework Convention on Climate Change (UNFCCC). I think it's important to note that there is potential for double counting if REDD+ used for voluntary carbon markets and contribution to country's NDC.	Thank you. We have re-written parts of the Introduction (Lines 40-53) to make clearer the inter-relationships between INDC's and REDD+. We also included text to emphasize the current discussions in the UNFCCC concerning the double counting of carbon credits. (Lines 290-294)
The paper is written like policy driving research, but I prefer the other way around – state importance first (contribution of mangroves to national carbon stocks), then place in context of policy needs to preserve the carbon stocks.	Thank you. We appreciate the reviewer's point but we respectfully opted to give a stronger policy content to this work which will be of interest to the Nature Communications readership. This choice is based on: 1. There are a number of papers describing carbon stock estimates from global and Brazilian mangroves, and yet, the Brazilian government has not moved forward in implementing any mitigation or adopted necessary steps for their inclusion into REDD+ financing; 2. We also included a section showcasing the well-established stocks found in this work, with a brief discussion on their regional variability in a global context (Lines 207-231). As such we believe we have addressed this reviewer's comment, where their importance to carbon sequestration is noted.
State in the introduction, what carbon stocks were included in the TECS -above-ground biomass, below-ground biomass, downed wood, and soil carbon?	Thank you, now included in introduction (Lines 63-66)
107 – 21% less carbon than what?	Thank you, the information was added as requested (Lines 112-115) "Mangroves occurring on open-coast hydrogeomorphic settings have 21% less carbon when compared to estuarine and delta settings..."
Fig 2 – need to spell TAGC and TBGC out in the caption. Can't see the other half of the error bar.	Thank you, this figure was updated and now shows the appropriate captions and error bars
107-126 – no mention of the other carbon stocks and variation between hydrogeomorphic settings, but looks like downed wood is greatest in delta open coast, and BGC is proportional to ABC.	Downed wood was higher on estuarine mangroves, we added this info in the results. Lines 124-126
147 – The mangroves of concern are unclear in this sentence: "Mangrove TECS in the Amazon	Thank you. We corrected this sentence (Lines 154-155) "The TECS loss of mangrove converted areas (shrimp ponds and

averaged 351.2 Mg C ha⁻¹ as a result of their conversion to shrimp farms and pastures.”	pastures) in the Amazon averaged 351 Mg C ha⁻¹”
174-176 – Is this estimate of avoided emissions based on the 2008-2013 or the 2016-2021 deforestation rates. Seems it would be an overestimate if based on the earlier. FREL for mangrove should be based on the later deforestation rate, given deforestation rates have declined.	Thank you. We corrected the avoided emissions estimate based on a FREL from 2016-2021 (Supplementary Table 8) and Lines 178-179
175-176 – it is not the mangrove loss that would generate carbon credits, but avoided loss of mangroves from land-use change. And please clarify in the introduction how carbon credits are sold under the REDD+ framework.	Thank you, corrected the sentence (Lines 190-194) to : “Considering a best-case scenario, actions to avoid mangrove loss from land-use change in the Legal Amazon could potentially generate nearly 9.2 ± 0.11 Mt CO₂e over a period of 10-yr under the REDD+ framework....”
190 – While the forest stock assessment is very comprehensive, I wouldn’t agree that the land use change assessment is that comprehensive (4 shrimp ponds and 1 pasture). For ponds is acceptable, but wouldn’t capture any variation for the pasture.	Thank you, we removed the “land use term” from this sentence. As indicated above, we now highlight the limitation on the replication of pastures given the limited access to private land. We included a new paragraph in the discussion (Lines 281-294) where we compared our emission factors with a global review, and with land-use emission factors from pastures in upland forests. Our emission factors (71-83% of loss in C stocks) are well within the range of other global datasets (mean of 82%, Sasmito et al., 2019 Global Change Biology).
199 -201 Inclusion of other environmental variables in the analysis would make this assessment more robust and would confirm the influence of other environmental, climatic and physiographic factors, as you say these were important predictors of the amount of carbon stored in mangrove forests other than hydro-geomorphological settings in the introduction. For example, you could include species composition, salinity, bulk density, sedimentation/erosion, rainfall, temperature. The role of other environmental variables is only briefly discussed and could be enhanced with citing other studies.	Lines 127-136 - We agree and we used multiple linear regression to test for the association of climatic (precipitation), hydrological (salinity), latitude and tree basal area with total ecosystem carbon stocks. We included these variables as they could be easily obtained by weather stations, simple field measurements or satellite observations. After removing significant co-variables, we show that tree basal area had a significant, but limited power to predict TECS (Adjusted R²= 0.30). We then used the Akaike Information Criterion (AIC) to select the best model

	to predict TECS in the Amazon coast, and obtained a reduced model with soil salinity and tree basal area with a slightly better Adj R ² =0.36. We have detailed in the manuscript these results and discuss these patterns in light of the global literature.
239 – 240 – You mean efficient in terms of per unit area. However, terrestrial ecosystems cover a greater area, so they may still offer similar avoided emissions, if deforestation rates are similar.	Thank you. Lines 295-297 - We have re-written this sentence to: “Similar to mangroves throughout the world, Amazon mangroves are far more efficient offsetting emissions, in terms of per unit area, when compared to terrestrial forests, savannas and grasslands of Brazil.”
241 – State what the potential annual mangrove mitigation is – avoided clearing in XXXX ha of mangroves. And is this equivalent to the carbon accumulation in 300,000 ha of secondary forests, also annually?	Thank you, we have included this information. Lines 297-299. “For example, the potential annual mangrove mitigation in the Legal Amazon (751 ha yr⁻¹) is equivalent to the annual net carbon accumulation in 82,400 ha of secondary forests⁴².”
Mention the stable isotope analysis in the results, but don’t elaborate in the discussion, whether the higher soil carbon stocks in estuary mangroves has something to do with the source of carbon, e.g. because the estuaries have both freshwater allochthonous carbon and mangrove autochthonous carbon, whereas the deltas have mainly allochthonous?	Thank you, we agree. We have elaborated further in the discussion the implications of allochthonous C for mitigation purposes – Lines 222-231
Methods – what is a “delta/open coast” setting?	We used this term to indicate that our open coast sites are in the Amazon delta
Only assessed land use effects in estuary mangroves, which have higher soil carbon than the other geomorphic settings, so applying the estuary emission factor to the potential loss of all mangroves across the Amazon is likely to be an overestimate, because some of this deforestation would be in delta and open coast mangroves, where the EF’s are likely to be lower, because of lower initial soil carbon stocks. How do the EFs developed compared to other studies, e.g. Sasmito et al. 2019 https://doi.org/10.1111/gcb.14774	Thank you. Lines 281-294 - Indeed, if we only applied the conversion to any land-use area in the Amazon there would be some uncertainties on emissions in areas with lower C stocks. We included a paragraph in the discussion pointing out these uncertainties, as suggested by the other reviewer, where we compare the emission factors from our sites with Sasmito et al., 2019 work and others. We discuss that our estimates are well within the range of global estimates for loss of C in land-use settings, and even comparable to upland emissions from pasture conversion. We included one of the tables (now Table 3) that was in the

	Suppl. Material into the text to make this point clear.
At each site, six plots were established, which is great replication. But why were plots established along a 100 m transect in a perpendicular direction from the mangrove/estuary or coast? That would capture zonation across the transect. Wouldn't it be better to establish parallel, then place the six plots to represent the different zones of mangroves, associated with species and structure? Please explain why you did the field design this way and any implications to the results.	The reviewer is correct. The placement of transects perpendicular to the marine-mangrove ecotone is designed to capture the zonation between the marine – forest variability. We can't say if it would be better to place plots in parallel to the rivers or to the coast, but it is likely that we would miss information about the zonation. For that reason, we decided to follow the Kauffman and Donato, 2012 protocol, which is widely used. We do not believe the results would suffer from any methodological artifacts by our choice.
Reviewer 3	
The manuscript titled “The inclusion of Amazon mangroves in Brazil’s REDD+ program” is a good quality paper which highlighted how can we convert the carbon stock data from mangroves to REDD+ which is globally significant. The research was done in an important mangrove ecosystem of the world and used 196 plot data. This is very appreciating. The authors had done a tremendous field work to obtained the data.	We thank the reviewer for this comment
1. In many areas it is written that carbon stock can be used for mitigating climate change. The total ecosystem carbon stock assessment actually depicts vulnerability potential to climate change in the case of its deforestation. The carbon accumulation rate per year actually will estimate its mitigation potential to climate change. Even a high total ecosystem stocked areas may be having low carbon accumulation rate due to LULCC especially when it is converted to aquaculture farm. Therefore, TECS assessment not alone can't be used for climate change mitigation. However, will be a good scientific document for REDD+ funding. The authors can rewrite and can emphasis on the importance of carbon accumulation rate assessment also for the accurate carbon credits. Refer : “Relevance and magnitude of 'Blue Carbon' storage in mangrove sediments: Carbon accumulation rates vs. stocks, sources vs.	Thank you, we appreciate the reviewer's comment. We have included a sentence to stress the differences among these concepts and have included the suggested citation in the text. Lines 54-60 We added a sentence on the discussion to stress the importance of C accumulation estimates to assess mangrove mitigation potential – Lines 222-231

sinks". https://doi.org/10.1016/j.ecss.2020.107027	
2. Also please confirm whether Tier 3 method is for carbon stock change over a period of time or just for carbon stock. The description of the methodology is not clear whether they used time series data or not?	Correct – tier 3 refers to change over a period of time. We corrected the term to Tier 2, indicating a regional assessment of carbon stocks without a time-series assessment. Lines 239
3. Also refer IPCC 2019 refinement to 2006 guidelines if any changes are there for the methodology.	We carefully reviewed the 2019 IPCC guidelines with respect to blue carbon ecosystems and to our knowledge, there were no refinements or updates for the methods/data that we used in this study.
Minor corrections	
1. In Fig.2 mark the label as Land use change instead of land use also paired or pristine mangrove. Please define the classification properly.	Thank you, this figure was corrected
2. Line no 154 The carbon emissions from mangroves in the Legal Amazon is 3-20-fold higher than land use emissions	Thank you, corrected Lines 161-163
3. Line no.153 CO2 is not the single GHG gas. CH4 emission is also very potential emission due to this aquaculture conversion. Therefore, rewrite the sentence.	Thank you, and we agree. We removed the term “carbon” emissions - Lines 161-163

REVIEWERS' COMMENTS

Reviewer #2 (Remarks to the Author):

The authors have been very detailed in addressing my comments and I support the publication of this important work.

Reviewer #3 (Remarks to the Author):

The manuscript revised well. However the Line no 57 can be more clear by adding carbon stocks instead of simply carbon.
possibly you can rewrite it as "The loss of mangrove forests have been shown to release massive amounts of carbon stocks that have been sequestered for centuries in deep soils and even in biomass".

Responses to the second round of reviews

Reviewers comment	Authors response
Reviewer #1	Thank you. We appreciate you for your time reviewing this manuscript and for your comments.
Reviewer #2 (Remarks to the Author)	
The authors have been very detailed in addressing my comments and I support the publication of this important work.	Thank you, the revisions were accepted and corrected.
Reviewer #3 (Remarks to the Author):	
The manuscript revised well. However the Line no 57 can be more clear by adding carbon stocks instead of simply carbon. Possibly you can rewrite it as "The loss of mangrove forests have been shown to release massive amounts of carbon stocks that have been sequestered for centuries in deep soils and even in biomass".	Thank you. Corrected as suggested by the reviewer – Line 58.